# Mental Health Challenges and Needs among Sexual and Gender Minority People in Western Kenya

**DOI:** 10.3390/ijerph18031311

**Published:** 2021-02-01

**Authors:** Gary W. Harper, Jessica Crawford, Katherine Lewis, Caroline Rucah Mwochi, Gabriel Johnson, Cecil Okoth, Laura Jadwin-Cakmak, Daniel Peter Onyango, Manasi Kumar, Bianca D.M. Wilson

**Affiliations:** 1Department of Health Behavior and Health Education, School of Public Health, University of Michigan, Ann Arbor, MI 48109, USA; jnicolem@umich.edu (J.C.); katlew@umich.edu (K.L.); gljohns@umich.edu (G.J.); ljadwin@umich.edu (L.J.-C.); 2Western Kenya LBQT Feminist Forum, Kisumu 40100, Kenya; rucahwarren@yahoo.com; 3Nyanza Rift Valley and Western Kenya (NYARWEK) LGBTI Coalition, Kisumu 40100, Kenya; conchela12@gmail.com (C.O.); muksdan2010@gmail.com (D.P.O.); 4Department of Psychiatry, School of Medicine, University of Nairobi, Nairobi 00100, Kenya; mkumar@uonbi.ac.ke; 5The Williams Institute, School of Law, University of California Los Angeles, Los Angeles, CA 90095, USA; WILSONB@law.ucla.edu

**Keywords:** Kenya, mental health, sexual and gender minority, LGBTQ, violence

## Abstract

*Background*: Sexual and gender minority (SGM) people in Kenya face pervasive socio-cultural and structural discrimination. Persistent stress stemming from anti-SGM stigma and prejudice may place SGM individuals at increased risk for negative mental health outcomes. This study explored experiences with violence (intimate partner violence and SGM-based violence), mental health outcomes (psychological distress, PTSD symptoms, and depressive symptoms), alcohol and other substance use, and prioritization of community needs among SGM adults in Western Kenya. *Methods*: This study was conducted by members of a collaborative research partnership between a U.S. academic institution and a Kenyan LGBTQ civil society organization (CSO). A convenience sample of 527 SGM adults (92.7% ages 18–34) was recruited from community venues to complete a cross-sectional survey either on paper or through an online secure platform. *Results*: For comparative analytic purposes, three sexual orientation and gender identity (SOGI) groups were created: (1) cisgender sexual minority women (SMW; 24.9%), (2) cisgender sexual minority men (SMM; 63.8%), and (3) gender minority individuals (GMI; 11.4%). Overall, 11.7% of participants reported clinically significant levels of psychological distress, 53.2% reported clinically significant levels of post-traumatic stress disorder (PTSD) symptoms, and 26.1% reported clinically significant levels of depressive symptoms. No statistically significant differences in clinical levels of these mental health concerns were detected across SOGI groups. Overall, 76.2% of participants reported ever using alcohol, 45.6% home brew, 43.5% tobacco, 39.1% marijuana, and 27.7% miraa or khat. Statistically significant SOGI group differences on potentially problematic substance use revealed that GMI participants were less likely to use alcohol and tobacco daily; and SMM participants were more likely to use marijuana daily. Lifetime intimate partner violence (IPV) was reported by 42.5% of participants, and lifetime SGM-based violence (SGMV) was reported by 43.4%. GMI participants were more likely than other SOGI groups to have experienced both IPV and SGMV. Participants who experienced SGMV had significantly higher rates of clinically significant depressive and PTSD symptoms. *Conclusions*: Despite current resilience demonstrated by SGM adults in Kenya, there is an urgent need to develop and deliver culturally appropriate mental health services for this population. Given the pervasiveness of anti-SGM violence, services should be provided using trauma-informed principles, and be sensitive to the lived experiences of SGM adults in Kenya. Community and policy levels interventions are needed to decrease SGM-based stigma and violence, increase SGM visibility and acceptance, and create safe and affirming venues for mental health care. Political prioritization of SGM mental health is needed for sustainable change.

## 1. Background

Kenya is home to an increasingly more visible sexual and gender minority (SGM) community, as can be seen by the growing number of civil society organizations (CSOs) and national advocacy efforts focused on improving the health and human rights of lesbian, gay, bisexual, transgender, and queer (LGBTQ) people [1]. This activism and advocacy is especially important considering the non-affirming, and in many ways hostile, social and public policy environment for LGBTQ people in Kenya [2]. This hostility is exemplified by the Kenya High Court’s decision on 24 May 2019 to reject a petition filed in 2016 by three LGBTQ-focused CSOs in Kenya that would have declared Sections 162 and 165 of the Kenya Penal Code unconstitutional, thus decriminalizing same-sex behavior among consenting adults. Currently in Kenya, consensual same-sex behavior is punishable by up to 14 years in jail [3]. Within this relatively anti-LGBTQ social and legal context, many SGM people in Kenya experience regularly documented human rights violations such as physical assault from mobs and vigilantes, rape and sexual assault by police, and institutional barriers to housing, education, and employment [2,4,5,6,7,8]. Further, research has shown that SGM people report experiencing harassment and denial of care from health care workers, and therefore also report frequently avoiding seeking physical and mental health services for fear of discrimination and even violence [9,10]. 

Much of the current stigma and discrimination experienced by SGM people throughout Sub-Saharan Africa stems from colonial laws that criminalize same-sex behaviors and diverse gender identities, often supported by Western religions that were also an import from European colonizers [1,11,12]. There is substantial historical evidence that prior to colonization by Western countries; there were multiple forms of same-sex behaviors, identities and relationships, as well as various expressions of gender identities across the African continent [13,14,15]. It is important to note that these practices and identities were not necessarily labelled as LGBTQ, as this nomenclature is often considered to have its origins in Eurocentrism [16,17]. Specifically in Kenya, multiple scholars and activists have challenged the notion that same-sex behaviors and identities are in opposition to traditional Kenyan culture and norms, citing historical examples of culturally sanctioned same-sex relationships in some Kenyan tribes, and calling for the rejection of anti-SGM laws which were created by oppressive colonizers, but have been maintained post-colonization [1,7,18,19,20]. Despite these challenges, many SGM people in Kenya experience pervasive violations of their human rights [2,4,5,6,7,8].

In addition to the direct material and physical impacts of stigma-related violence, many scholars have identified the mental health effects of prejudice and stigma related to sexual and gender minority status. The most empirically validated conceptual model making this connection is the Minority Stress Model, which posits that exposure to persistent stress stemming from anti-SGM prejudice, stigma, and discrimination places SGM individuals at increased risk for negative mental health outcomes, including higher rates of mental disorders and psychological distress, and lowered psychological and social wellbeing [21,22]. This model extends extensive transcontinental theory and empirical research with an array of populations and ages demonstrating that environmental adversity and stress are critical factors in the development of psychopathology and focuses on the unique experiences of sexual and gender minority people [23,24,25,26,27]. 

The forms of stress emphasized by the Minority Stress Model include major life events, such as assault because of their sexual and gender minority status, as well as everyday forms of discrimination, such as receiving poor services. Although the initial development and empirical research supporting the Minority Stress Model was conducted in the United States, the utility of this model has been demonstrated globally with populations of SGM individuals across six continents (c.f. [28,29,30,31,32,33,34,35]); including quantitative and qualitative studies with sexual minority men in Nigeria [35,36], South Africa [37,38], and Zambia [39]. For SGM people in Kenya and other resource-poor settings, the Minority Stress Model is particularly helpful in understanding mental health outcomes since it also recognizes that SGM minority status is situated within general environmental circumstances (e.g., poverty); resulting in the experience of general stressors (e.g., unemployment) as well as embedded sexual and gender minority stress processes at both the societal and individual levels [22,40]. In order to develop public policy efforts aimed at improving the mental health and wellbeing of SGM people in Kenya, research is needed regarding experiences of discrimination and trauma, and subsequent mental health concerns within this population. 

Limited national data exist on the mental health of LGBTQ people in Kenya, and the scant existing research has primarily been conducted with gay and bisexual men and other men who have sex with men (GBMSM) and is typically conducted within HIV research projects [41]. In addition to not being representative of non-GBMSM members of the Kenyan SGM community such as lesbian women and transgender individuals, this research often includes samples that consist predominately of GBMSM who engage in sex work or transactional sex (due to greater ease of recruitment); thus these limited data are likely not representative of the broader Kenyan SGM community, or even to the general GBMSM population in Kenya.

The extant data on GBMSM mental health in Kenya has documented relatively high rates of violence and traumatic incidents as well as mental health concerns and substance use. Kunzweiler et al. found that among 711 GBMSM in Western Kenya, 11.4% reported moderately severe or severe depressive symptoms, 50.1% reported harmful alcohol abuse, and 23.8% reported moderate substance abuse. In addition, this study found high rates of childhood physical or sexual abuse (80.9%) and recent physical or psychological trauma linked to anti-LGBTQ stigma (39.1%) [6]. These results are in line with findings from coastal Kenya, where one third of 112 GBMSM met criteria for major depressive disorder, 45% reported alcohol abuse, and 59.8% reported other substance abuse [4]. Additionally, 67% of participants reported some form of abuse within the past year and 77% reported some form of abuse in childhood [4]. A secondary analysis of data by Korhonen et al. from 1476 GBMSM abstracted from three HIV-focused studies conducted in the capital city of Nairobi, coastal Kenya, and Western Kenya (three regions containing the majority of Kenya’s population) found that 31% reported moderate-to-severe depressive symptoms, 44% reported hazardous alcohol use, and 51% reported problematic substance use—all higher than Kenyan national rates [5].

High rates of childhood abuse and recent experiences of anti-LGBTQ physical and psychological violence among GBMSM in Kenya have been repeatedly reported in the existing literature and provide additional support for examining the health of the Kenyan LGBTQ community through a Minority Stress lens [4,5,6]. All three of the studies on Kenyan GBMSM discussed above found significant associations between abuse or trauma and depressive symptoms, alcohol abuse, and other substance abuse. Such associations between trauma and symptoms of depression and anxiety have also been found in samples of young Kenyan GBMSM (ages 18–29) [42,43]. 

Limited data are available on the health-related experiences of lesbian, bisexual and queer women in Kenya [44]. Wilson et al. conducted an exploratory study that included an analysis of comments made during a facilitated community forum and an examination of the sociopolitical and legal environment relevant to sexual minority women’s health in Kenya. Participants identified multiple health-related issues, including concerns related to healthcare access, healthy sexual relationships, economic instability, and freedom from violence. The legal and policy analysis indicated that policy is complicated by the presence of hostile laws regarding same-sex sexuality, an absence of economic policies to protect women, and health policies that render sexual minority women invisible [2]. Another quantitative study of 273 sexual and gender minorities assigned female at birth in Western Kenya found that overall 27.7% of the participants experienced violence due to their sexual orientation, gender identity or gender expression, with those whose gender expression was masculine, androgynous/all-gender or who did not use a gender expression or role term were more than two times more likely to report such violence [45]. Of those who experienced SGM-related violence, 44% experienced verbal violence, 38.7% experienced emotional violence, 21.3% experienced physical violence, and 14.7% experienced sexual violence. The majority (55.8%) reported two or more forms of violence [2,45]. 

The burden of mental health and substance use disorders in Sub-Saharan Africa is projected to increase by 130% between 2010 and 2050 [46]. Given their status as a highly stigmatized minority group, this projected increase places Kenyan SGM people in a uniquely dangerous position. Interventions at multiple socio-ecological levels will be needed to dismantle the persistent interpersonal and institutionalized violence and discrimination that the community faces, and data are critically needed to more fully understand the situation and needs of this diverse population. In order to expand on the limited existing literature and to provide data on the broader SGM community in Kenya, this paper presents data from a sample of 527 SGM adults in Western Kenya. In line with the principles of the Minority Stress Model, which connects experiences of stigma and discrimination to mental and physical health outcomes, we examine factors such as experiences with violence (intimate partner violence and SGM-based violence), mental health outcomes (psychological distress, PTSD symptoms, and depressive symptoms), and alcohol and other substance use. We also examine the prioritization of needs, as described by the community members who participated in the study.

## 2. Materials and Methods

### 2.1. Study Design and Sampling

This study was designed and conducted by members of a collaborative partnership between a U.S. academic institution and a Kenyan CSO focused on SGM human rights. Members of the Kenyan CSO, all of whom identified as LGBTQ, conducted study recruitment, enrollment and data collection activities. The CSO is a network that, at the time of data collection, included 17 member organizations located throughout nine counties in Western Kenya. These organizations were divided into seven geographically close clusters to facilitate regional meetings and activities. Recruitment occurred through a two-stage process. In the first stage, information regarding the upcoming survey was shared with members of the broader SGM community in Western Kenya through social media announcements on the CSO’s Facebook page and in WhatsApp groups focused on LGBTQ people in Western Kenya. This initial sensitization also occurred through verbal discussions and announcements at member organizations’ group activities and events, and at social gathering spots such as bars and discos.

In the second stage, the same communication channels used during the study sensitization stage were used to recruit and enroll participants. Individuals were eligible to participate in this cross-sectional study if they were 18 years or older, lived in Western Kenya, were able to read English, and identified as a member of the LGBTQ community. During this second stage, outreach workers also conducted workshops on sexual and reproductive health rights in all of the cluster regions, and following those meetings any interested individuals were able to talk with one of the outreach workers in order to be screened and enrolled.

Participants had the option of responding to the survey on a paper questionnaire or online through a virtual and secure Qualtrics platform. All surveys were presented and completed in English. The measures in the survey were selected by members of the collaborative research team based on four focus groups that were conducted with members of SGM CSOs in Western Kenya to understand participants’ mental health conceptualizations, concerns, and challenges. The final survey included items to assess demographics (e.g., gender identity, sexual orientation, age, county of residence, religion, employment, educational level), violence (intimate partner and SGM), and the mental health areas of psychological distress, depressive symptoms, post-traumatic stress symptoms, and alcohol and substance use. All items on the survey were reviewed for comprehension and cultural appropriateness by the Kenyan CSO, and then pilot-tested with a sample of ten SGM participants from Kisumu. Based on pilot testing, the Kenyan CSO altered the wording on six of eighteen symptom description items in the Brief Symptom Inventory-18 to enhance local understanding and clarity.

#### 2.1.1. Sample General Demographics

A total of 570 participants completed the survey. Forty-three surveys were excluded from this analysis due to incomplete data and/or identification as cisgender and heterosexual, resulting in a final sample of 527 SGM adults. More than 90% of the participants were between the ages of 18–34 (Table 1). All of the participants resided in the westernmost counties in Kenya, with the vast majority residing in Kisumu (58.1%), Siaya (13%), Bungoma (9%), or Kakamega (6.5%). Participants reported their religion as predominately Catholic (37.8%), Anglican (21.5%), Seventh Day Adventist (14.8%), or Muslim (12%). Of the individuals who indicated their current employment status, almost half were employed either full time or part time (44.8%) while others were working as a laborer (9.4%), sex worker (10.6%), or were not working but in school (15.4%). The vast majority of the sample (82.5%) had either completed primary school, secondary school, or received their diploma. Very few had received their bachelor’s (5.5%) or master’s degree (0.6%). See Table 1 for more detailed descriptive demographics.

#### 2.1.2. Sample Sexual Orientation and Gender Identity (SOGI)

Participants reported a range of sexual orientation identities, with the vast majority identifying as bisexual (47.4%), gay (23.9%), or lesbian (13.3%). The remaining identified their sexual orientation as men who have sex with men/MSM (10.1%), women who have sex with women/WSW (2.1%), other (1.7%) or queer (1.3%). Among gender minority individuals, the fifty-seven people who indicated they were transgender reported their sexual orientation as bisexual (19.3%), gay (36.8%), MSM (21.1%), lesbian (14%), other (5.3%), or queer (1.8%). Only one participant who identified as a transgender man indicated they were heterosexual. Genderqueer or gender non-conforming people specified their sexual orientation as either bisexual (60%) or queer (40%).

Participants were asked to identify their sex assigned at birth and their current gender identity. Seventy-two percent were assigned male sex at birth and 28% were assigned female sex. Based on participants’ responses to the sex assigned at birth and current gender identity questions, we created three unique gender identity groups: (1) cisgender men and women (individuals whose current gender identity is the same as their sex assigned at birth (88.2%), (2) transgender men and women (individuals whose current gender identity is different than their sex assigned at birth (10.8%), and (3) gender non-binary (individuals whose current gender identity is genderqueer or gender non-conforming regardless of their sex assigned at birth (1.0%). See Table 2 for more detailed sample sexual orientation and gender identity data.

### 2.2. Measures

#### 2.2.1. Demographic Information

Demographic information was assessed using questions developed by the collaborative partnership. Demographic information regarding county of residence, religion, employment, and educational level were assessed using multiple-choice questions that included response options previously developed by the Kenyan CSO for prior surveys. Age was asked as an open-ended question. Gender identity was assessed using a two-step method (Step 1: current gender identity, Step 2: assigned birth sex) that has been used globally to assess transgender identity [47]. Sexual orientation was assessed using a multiple-choice question with the following potential response options, which were used by the Kenyan CSO for prior surveys: bisexual, lesbian, gay, queer, straight/heterosexual, MSM (man who has sex with men), WSW (woman who has sex with women), and other.

#### 2.2.2. Psychological Distress

Psychological distress was assessed using the Brief Symptom Inventory-18 (BSI-18) [48], an 18-item measure that has been successfully used to assess psychological distress symptoms among adults in Western Kenya, and elsewhere in Sub-Saharan Africa [49,50,51]. Based on pilot testing with SGM adults in Western Kenya, wording on six of the eighteen symptom items was altered to be in alignment with local understanding of these symptoms (e.g., “feeling blue” was changed to “feeling low” and “spells of terror or panic” was changed to “moments of fear or panic”). Psychological distress items were categorized into three areas, somatization, depression, and anxiety based on a scale of 0 = “not at all likely” to 4 = “extremely likely.” Here, we report specifically on the global score index (GSI) of all BSI scores by computing the z-score and then translating it to a *t*-score where two groups were created based on a score greater or less than 62, with a *t*-score greater than 62 indicating clinical significance. If item values were missing for more than three items, the score was considered invalid. The 18-item scale was found to be highly reliable (α = 0.94).

#### 2.2.3. Depressive Symptoms

Depressive symptoms were assessed using the Patient Health Questionnaire (PHQ-9), a 9-item measure that has been successfully used to assess depressive symptoms among GBMSM adults in Western Kenya, and elsewhere in Sub-Saharan Africa [5,52,53,54]. A total continuous score was calculated by adding together responses from all nine items, which required frequency responses ranging from 0 = “not at all” to 3 = “nearly every day.” Means were imputed for 1–2 missing items and those missing >2 items did not receive a score. Scores were then categorized as follows: 0–4 (no to minimal depression), 5 to 9 (mild depression), 10 to 14 (moderate depression), 15 to 19 (moderately severe depression), and 20–27 (severe depression). Categories were dichotomized by not clinically significant levels of depressive symptoms (no to mild depression; ≤9) and clinically significant levels of depressive symptom warranting clinical attention (moderate to severe depression; ≥10). The 9-item scale was found to be highly reliable (α = 0.88).

#### 2.2.4. Post-Traumatic Stress Disorder Symptoms

Post-traumatic stress disorder symptoms were assessed using the Primary Care Post-Traumatic Stress Disorder (PC-PTSD) Assessment, a 4-item measure that has been successfully used to assess PTSD symptoms among adults in Kenya, and elsewhere in Sub-Saharan Africa [55,56,57]. Items were summed and categorized on a scale of 0–4, with 3–4 indicating the need for a clinical evaluation for PTSD, and 0–2 indicating that clinical evaluation was not needed. The 4-item scale was found to be highly reliable (α = 0.88).

#### 2.2.5. Alcohol and Substance Use

The use of alcohol and other substances was assessed using items from the Kenya National Authority for the Campaign against Alcohol and Drug Abuse (NACADA) Rapid Situation Assessment [58]. We focused on the use of alcohol, home brew (local home-brewed spirits), tobacco, marijuana, and miraa or khat (local stimulants). These five different types of alcohol and substance use were dichotomized as (1) ever used the specific type of alcohol/substance, and (2) never used the specific type of alcohol/substance. In addition to the first type of dichotomous variable, we also created another dichotomous variable to determine the potential presence of problematic alcohol/substance use, which we defined as daily use.

#### 2.2.6. Intimate Partner Violence and Sexual/Gender Minority-Based Violence

Lifetime experiences of Intimate Partner Violence (IPV) and Sexual/Gender Minority-Based Violence (SGMV) were assessed with questions developed by the Kenyan CSO based on a prior community needs assessment. For IPV, participants were asked “Have you ever been the victim of violence from a current or past intimate partner?” and if they responded affirmatively, follow-up questions assessed the type(s) of violence experienced and the perpetrator(s) of the most recent episode of violence. For SGMV, participants were asked “Have you ever been the victim of violence because of your sexual orientation, gender identity, or gender expression (like, how womanly or manly you seem to others)?” and if they responded affirmatively, follow-up questions assessed if the violence was because of their sexual orientation and/or gender identity/gender expression, the type(s) of violence experienced, and the perpetrator(s) of the most recent episode of violence.

### 2.3. Statistical Analyses

We calculated descriptive frequencies and means for all variables of interest, and then for comparative analytic purposes, we created three groups based on participants’ sexual orientation and gender identity (SOGI). This included the following three groups: (1) cisgender sexual minority women (SMW; cisgender women who identified as any non-heterosexual identity, *n* = 131, 24.9%), (2) cisgender sexual minority men (SMM; cisgender men who identified as any non-heterosexual identity, *n* = 336, 63.8%), and (3) gender minority individuals (GMI, transgender and gender non-binary regardless of their sexual orientation; *n* = 60, 11.3%).

We conducted chi-square tests for independence to test for the association between SOGI group and scoring on the following three measures of mental health concerns: psychological distress, PTSD symptoms and depressive symptoms. In order to dichotomize these three mental health measures, we used the standard clinical cut-off scores associated with each standardized measure [53,59,60] to create two groups for each—those with clinically significant levels of symptoms and those without. We also conducted chi-square tests for independence to compare scores on the alcohol and substance use measures across the three SOGI groups. For these variables, we dichotomized scores based on whether or not the participant reported potentially problematic use, which we operationalized as daily alcohol and/or substance use.

We conducted another set of chi-square tests for independence to examine the association between lifetime experiences of Intimate Partner Violence (IPV) and Sexual/Gender Minority-Based Violence (SGMV) and SOGI group, and compared their standardized residual values. In addition, we used chi-square tests to examine potential associations between lifetime experiences of SGMV and clinical cut-off scores on our mental health variables (psychological distress, PTSD symptoms and depressive symptoms). Across all chi-square tests of independence, we calculated standardized residual values for each cell as an indicator of effect size, and to determine which cells were contributing the most to the chi-square value.

## 3. Results

### 3.1. Psychological Distress, PTSD, and Depressive Symptoms 

Overall, 11.7% of the participants reported clinically significant levels of psychological distress (average *t*-score > 62 was 50.2, SD 10.09), as measured by the BSI-18 Global Severity Index. A comparison of the SOGI categories indicated that GMI and SMM had nearly identical levels of clinically significant psychological distress (12.7% vs. 12.3%, Table 3) indicating a score of 62 or more. They were followed by SMW who had lower levels of clinically significant psychological distress (9.5%), however these differences were not statistically significant. More than half (53.2%) of participants reported clinically significant levels of PTSD symptoms indicative of a probable diagnosis of PTSD (endorsing three or more items on the PC-PTSD). A comparison of the three SOGI categories indicated no statistically significant differences in PTSD symptom severity, although there was a slightly higher proportion of GMI who reported more severe PTSD symptoms indicating the need for clinical attention. See Table 3 for comparisons of psychological distress and PTSD symptom scores across the three SOGI categories.

The overall averaged score for the PHQ-9 scale was 7.37, SD 6.09. A little over one third of the participants (35.7%) reported minimal or no levels of depressive symptoms, and 38.2% reported mild levels of depressive symptoms; thus approximately three quarters (73.9%) of the sample were likely not experiencing depressive symptoms at a severity level associated with a major depressive disorder (PHQ-9 ≤ 9). Moderate levels of depressive symptoms were reported by 13.1% of participants, while another 13.0% reported either moderately severe or severe levels of depressive symptoms, thus 26.1% of the sample reported symptoms that would warrant treatment for a major depressive disorder. A comparison of clinically significant levels of depressive symptoms (PHQ-9 score ≥ 10) across three SOGI categories indicated no statistically significant differences (SMW = 22.5%; SMM = 28.4%; GMI = 21.4%). A comparison of severe levels of depressive symptoms (PHQ-9 ≥ 20) across the three SOGI categories revealed statistically significant group differences, with GMI reporting the highest levels of severe depressive symptoms (GMI = 8.9%; SMM = 4.3%; SMW = 1.6%). See Table 3 for comparisons of depressive symptoms across the three SOGI categories.

### 3.2. Alcohol and Substance Use

Overall, 76.2% of participants reported having ever used alcohol (SMW = 63.6%; SMM = 80.6%; GMI = 80%); 45.6% home brew (SMW = 24.8%; SMM = 54.7%; GMI = 41.5%); 43.5% tobacco (SMW = 33.6%; SMM = 49.1%; GMI = 33.3%), 39.1% marijuana (SMW = 31.3%; SMM = 43.5%; GMI = 30.9%), and 27.7% miraa or khat (SMW = 16.5%; SMM = 30.6%; GMI = 37.0%). SMW had considerably lower than expected residual values for all sub-stances, whereas GMI were nearly comparable to SMM.

In terms of potentially problematic alcohol and substance use, we examined rates of daily use across all types of alcohol and substances: 17.1% tobacco, 16.9% alcohol, 9.9% marijuana, 5.2% home brew and 0.8% miraa or khat. In order to understand group differences in potentially problematic alcohol and substance use, we compared daily use among the three SOGI categories. The association between daily alcohol use and SOGI group was statistically significant (ӽ^2^ = 6.74, *p* = 0.034), and the standardized residuals indicate that the lower than expected levels of daily alcohol use among GMI contributed most to this association. A similar pattern was seen with daily tobacco use, where the association between daily tobacco use and SOGI group was statistically significant (ӽ^2^ = 12.78, *p* = 0.002), and the standardized residuals indicate that the lower than expected levels of daily tobacco use among GMI contributed most to the association. Examination of daily marijuana use showed a different pattern, whereby the association between daily marijuana use and SOGI group was statically significant (ӽ^2^ = 12.10, *p* = 0.002), but examination of the standardized residuals indicate that the higher than expected levels of daily marijuana use among SMM contributed most to the association. See Table 3 for comparisons of potentially problematic alcohol and substance use across the three SOGI categories.

### 3.3. Intimate Partner Violence (IPV) and Sexual/Gender Minority-Based Violence (SGMV)

Nearly half of participants indicated that they had ever experienced violence from an intimate partner (42.5%), and 43.4% reported ever experiencing discriminatory violence perpetrated against them because of their sexual orientation, gender identity and/or gender expression (i.e., Sexual/Gender Minority-Based Violence: SGMV). The association between ever experiencing IPV and SOGI group was statistically significant (ӽ^2^ = 19.061, *p* = 0.000), and the standardized residuals indicate that higher than expected levels of IPV among GMI contributed most to this association. A similar pattern was seen with SGMV, where the association between SGMV and SOGI group was statistically significant (ӽ^2^ = 43.917, *p* = 0.000), and the standardized residuals indicate that the higher than expected levels of SGMV among GMI contributed most to the association.

Among those who reported ever experiencing SGMV, 68.6% reported experiencing SGMV based on their sexual orientation, 25.5% based on their gender identity or expression, and 5.5% based on both their sexual orientation and gender identity or expression. Among the four types of SGMV assessed, 42.1% reported only experiencing verbal violence, 19.9% reported only experiencing physical violence, 16.9% reported only experiencing sexual violence, 5.6% reported only experiencing emotional violence, and 15.3% reported experiencing two of more of these types of violence. With regard to the perpetrator of the most recent SGMV incident, 31.6% of participants indicated that it was a former partner, boyfriend or ex-spouse, 18.2% indicated that it was a date, 13.3% indicated that it was a current spouse or partner, and 42.7% indicated that it was someone else (“other”). For the last category participants were asked to write in the type of person who perpetuated the last incident of SGMV they experienced and the following persons were indicated: friends (30.4%), parents/relatives (19.9%), stranger (19.3%), police (13.0%), healthcare provider (10.6%), religious leader (5.6%), government official (0.6%) and landlord (0.6%). See Table 3 for comparisons of rates of ever experiencing IPV and SGMV across the three SOGI categories.

### 3.4. Sexual/Gender Minority-Based Violence and Mental Health

In order to understand the mental health consequences of those experiencing SGMV, we compared scores for mental health variables among those who reported experiencing violence based on sexual orientation, gender identity, and/or gender expression, and those who did not report such violence. Those who experienced SGMV had significantly higher rates of moderate to severe depressive symptoms (35.0%) as compared to those who had not experienced such violence (18.6%; *p* = 0.000). Similarly, those who had experienced SGMV were more likely to report clinically significant levels of PTSD symptoms (67.3%) as compared to those who had not reported such violence (41.4%; *p* = 0.000). Psychological distress (indicated by a total BSI score > 62) occurred slightly more frequently among those who ever experienced SGMV than those who never experienced SGMV (13.5% vs. 9.9%) although this difference was not statistically significant (*p* = 0.233). See Table 4 for comparisons of mental health variables across SGMV categories. 

Of the 527 total respondents, 281 offered responses regarding the type of additional support and services they would like to receive. Two authors manually read all of the responses and initially categorized them into seventeen main themes, discussing discrepancies until they arrived at consensus. From these seventeen themes, the same authors further collapsed similar themes into seven broader thematic content areas representing the types of supports and services that participants stated they needed (not mutually exclusive). These included: Counseling and Mental Health Services (*n* = 64; 22.8%), Financial and Economic Empowerment (*n* = 55; 19.6%), Emotional/Peer Support (*n* = 42; 14.9%), Community Organization Support and Advocacy (*n* = 41; 14.6%), Healthcare Assistance and Medical Coverage (*n* = 39; 13.9%), Sexual Education and Prevention Items (*n* = 20; 7.1%), Nothing/Everything is Good (*n* = 20; 7.1%). Figure 1 displays the frequency with which participants expressed each content area.

## 4. Discussion

To our knowledge, this is the first published study that sought to document mental health challenges, experiences with violence, alcohol and other substance use, and stated needs among a sample of 527 SGM adults in Western Kenya. We also believe this to be the first combined sample of SGM individuals in Kenya with a focus on mental health and related factors. These data will be helpful in planning for future prevention and treatment efforts aimed at improving the health and wellbeing of SGM people in Kenya, as well as public policy focused on the health and human rights of this population. 

We examine our data through the lens of the Minority Stress Model, which connects experiences of stigma and discrimination to mental health outcomes for SGM people. This framing is important in the Kenyan context given that many SGM people experience high levels of stigma and human rights violations such as physical assault from mobs and vigilantes, rape and sexual assault by police, and institutional barriers to housing, education, and employment [2,3,4,5,6,7]. The Minority Stress Model is useful in understanding how exposure to persistent stress in the form of anti-LGBTQ prejudice, stigma, and discrimination can contribute to elevated rates of mental health challenges for SGM people in Kenya, as opposed to there being anything inherently dysfunctional about being a SGM person. Findings from our study now join prior investigations in Nigeria [35,36], South Africa [37,38], and Zambia [39] in supporting the utility of the Minority Stress Model in understanding the mental health challenges of SGM populations in Sub-Saharan Africa.

### 4.1. Psychological Distress and Post Traumatic Stress Symptoms

With regard to psychological distress, 11% of SGM adults in our sample reported clinically significant levels of psychological distress as measured by the BSI-18. This level is higher than that found among 397 adults living with HIV (ages 18–61; 71.5% women; SOGI not reported) in Western Kenya who were participating in psychosocial support groups in conjunction with their HIV treatment [49]. That study used the 53-item version of the BSI and found that only 3.5% scored in the clinical range on the BSI Global Severity Index. Both adults living with HIV and SGM adults are populations that suffer marginalization and oppression in Kenyan society, yet our sample demonstrated three times the rate of clinically significant psychological distress as was found among adults living with HIV. It may be that their sample was gaining benefits from psychosocial support groups and thus were experiencing less distress, or there may be measurement differences in the BSI-18 and BSI-53 versions of the scale or measurement challenges due to the cultural adjustments we made to wording in our BSI-18 measure.

Although the Kenya Mental Health Policy 2015–2030 identifies that there are inadequate data on the national prevalence of mental health, neurological, and substance use disorders in Kenya [61], the Ministry of Health estimates that 10% of the country’s general population suffers from a common mental disorder; and that this increases to 25% among people receiving routine outpatient health services [62]. Assuming this estimate is correct, the 11% of SGM people in our sample that reported clinically significant levels of psychological distress are in alignment with what has been estimated for mental health disorders among the general public. It is noteworthy though that the BSI-18 is a screening instrument assessing psychological distress, and not a diagnostic tool assessing mental disorders, thus caution is warranted in comparing psychological distress findings directly to rates of diagnosed mental disorders. Regardless, the rates of clinically significant psychological distress we found among SGM adults suggest that this population is experiencing mental health challenges and should be addressed, and potentially prioritized, by the Kenya Ministry of Health and in the Kenya Mental Health Policy 2015–2030.

More than half (52.2%) of SGM adults in our sample reported clinically significant levels of PTSD symptoms indicative of a probable diagnosis of PTSD. A comparison of the three SOGI categories indicated no statistically significant differences in PTSD symptom severity, although there was a slightly higher proportion of gender minority individuals with PTSD symptom profiles suggesting the need for clinical attention. These rates of probable PTSD are nearly 5 times higher than what was found among 1147 residents of the Maseno area in Kisumu County (the county of residence for 58.1% of our sample) who participated in a cross-sectional household survey [63]. This study found that 48% of participants had experienced a severe trauma, but the overall prevalence rate of probable PTSD was 10.6%, defined as a score of six or more on the Trauma Screening Questionnaire, and the conditional probability of PTSD was 26%.

A cross-sectional, national, population-based cluster survey of 956 Kenyan adults aged 18 and older used the same measure of PTSD as we did, and found that 33% of their sample reported clinically significant levels of PTSD [56]. The authors suggest that their data represent national PTSD symptom prevalence for adults in Kenya, noting the lack of national comparisons available. If their data are indicative of national levels of PTSD symptoms in the general population, our data suggest that SGM adults in Kenya represent a highly vulnerable population for PTSD. The Kenya Mental Health Policy 2015–2030 designates five groups as vulnerable to mental health conditions (i.e., children and adolescents, women, older persons, prisoners, people emerging from conflicts and disasters) and thus in need of targeted mental health interventions. The policy acknowledges that those engaged in conflict experience elevated stress and trauma that can lead to mental disorders. Our data suggest that SGM adults may fall within the group that experiences societal conflict, as evidenced by our data on both violence and PTSD, and thus warrant consideration as a vulnerable group in the Kenya Mental Health Policy.

### 4.2. Depressive Symptoms

Data on depressive symptoms using the PHQ-9 demonstrated that approximately three quarters (73.9%) of the sample were likely not experiencing depressive symptoms at a severity level associated with a major depressive disorder (PHQ-9 ≤ 9). Given the amount of stress and oppression members of this community face in Kenya, it is encouraging that this percentage is as high as it is. Despite this, 26.1% of our sample did report symptoms that would warrant treatment for a major depressive disorder. The only depression symptom difference among our three SOGI groups was with severe levels of depressive symptoms, whereby gender minority individuals reported the highest levels of severe depressive symptoms.

Our overall rate of clinical levels of depressive symptoms (26.1%) was lower than that found in the cross-sectional, national, population-based cluster survey of 956 Kenyan adults previously referenced, as they also used the PHQ-9 to measure depressive symptoms and found that 36.5% of their sample reported clinically significant levels of depressive symptoms [56]. As with PTSD, the authors suggest that their data represent national major depressive disorder symptom prevalence for adults in Kenya, noting the lack of national comparisons available. The lower rates of depressive symptoms found among our SGM participants may be a result of our sampling strategies. Since we relied heavily on member organizations affiliated with a regional Kenyan CSO focused on SGM health and human rights, it may be that participants were more likely to be those connected to SGM services and programs. These connections with other SGM individuals and organizations may have provided social support that buffered participants from experiencing more severe levels of depressive symptoms.

Three studies on GBMSM in Kenya also used the PHQ-9, and from the data reported in two of these studies it appears that our rates of not experiencing a major depressive disorder were slightly higher than a study on the coast (58%) but closer to the combined dataset from three regions in Kenya (69.4%) [4,5]. When examining elevated rates of depressive symptomatology, in the current sample we found that 13% of participants reported moderately severe or severe depressive symptoms (PHQ-9 ≥ 15), which was slighter higher than that found among GBMSM in a similar region in Western Kenya (10.5%) [64] and in the three-region dataset (12.2%) [5], but lower than that found among GBMSM on the Coast (23.2%) [4]. Examination of PHQ-9 data at these two ends of the depressive symptom continuum are clinically meaningful when using a brief tool such as the PHQ-9, as the original validation studies found that scores less than 10 seldom occurred in individuals with major depressive disorders while scores of 15 or greater usually signified the presence of a major depressive disorder [53]. In addition, the PHQ-9 has been used with various populations in Kenya and has been determined to be a reliable measure of depressive symptomatology [65].

### 4.3. Alcohol and Substance Use

With regard to alcohol and substance use, more than three quarters (76.2%) of the sample reported ever using alcohol, followed by home brew (45.6%), tobacco (43.5%), and marijuana (39.1%). Across the full sample, daily use was most common for alcohol (16.9%) and tobacco (17.1%), followed by marijuana (9.9%). Statistically significant SOGI group differences in daily use were revealed for these three substances, with GMI reporting significantly lower rates of daily use for alcohol and tobacco, and SMM reporting significantly higher rates of daily use for marijuana (as compared to the other groups). Across these three categories, SMM reported the highest levels of daily use (tobacco = 21.4%; alcohol = 19.4%; marijuana = 13.3%).

The elevated levels of alcohol and substance use we found among sexual minority men are in alignment with prior studies of GBMSM in Kenya, although these studies measured it using a more comprehensive questionnaire that assessed problematic alcohol and substance use. The GBMSM sample in Western Kenya revealed that nearly half (49.9%) of participants reported harmful alcohol use, whereas the three-region and coastal samples reported 44.0% and 44.6% respectively.

These rates are higher than those found among the 1147 participants in the cross-sectional household survey in the Maseno area in Kisumu County previously referenced, whereby prevalence of lifetime alcohol use for women was 6.8% and for men was 14.5%; and prevalence of hazardous alcohol use was 9.5% for men and 2.9% for women [66]. In a nationally representative household survey of 4203 adults aged 18–69 years conducted in Kenya, nearly 40% of respondents reported having ever consumed alcohol, 40.4% reported consuming alcohol within the past 7 days, and 12.7% reported heavy episodic drinking (six or more drinks on at least one single occasion per month) [66]. Our data, and those from GBMSM studies, illustrates that alcohol use, and potentially problematic alcohol use, appears to be elevated among SGM individuals in Kenya. This may be related to members of the SGM community finding access to others who share their sexual orientation and/or gender identity in social venues where alcohol is served, or to the use of alcohol as a way to feel more comfortable interacting with other LGBTQ people due to societal stigma and shame. It also may be that SGM individuals are using alcohol as a way to self-medicate against the pain and suffering they experience as a result of societal stigma and discrimination.

### 4.4. Intimate Partner Violence and SGM-Based Violence

Our study expanded the scope of data on violence-related data from those of GBMSM in Kenya by reporting data on lifetime experiences of violence related to sexual orientation, gender identity, and/or gender expression (SGM-based violence; SGMV), as well as intimate partner violence. We found that rates of lifetime SGMV (43.4%) and IPV (43.2%) were quite similar for the full sample, and that these rates were higher for gender minority individuals (SGMV = 84.2%; IPV = 69.9%). These higher rates of violence experienced by gender minority individuals is similar to data from 273 sexual and gender minorities assigned female at birth in Western Kenya where 27.7% of the participants experienced violence due to their sexual orientation, gender identity and/or gender expression, but these rates were two times higher (32.8% to 37.1%) among those whose gender expression was masculine (35.2%), androgynous/all-gender (37.1%) or who did not use a gender expression or role term (32.8%) [45].

Our rates of reported lifetime SGMV were comparable for cisgender sexual minority women (37.7%) and cisgender sexual minority men (38.3%). In addition, these rates were similar to the lower end of the range of recent physical or psychological trauma reported in the Anza Mapema study in Kisumu, where they found 39.1% of GBMSM in their mixed sample of both HIV-positive and HIV-negative participants reported such trauma, as did 51.5% of GBMSM in their HIV-negative sample [6,64]. Data from Coastal Kenya demonstrated slightly higher rates, with 66.9% of GBMSM reporting forced or coerced sex, physical abuse, emotional abuse, or threats or intimidation related to their same-sex behavior within the past year [4]. Korhonen et al. 3-site dataset of 1476 GBMSM found similar rates of recent trauma or abuse related to same-sex behavior, with a prevalence of 51.2% [5]. Our rates of SGM-based violence were generally lower and were based on lifetime violence vs. these other studies which focused on more recent violence (2 weeks to 1 year). Our lower rates of violence may be related to lower rates of sex work reported in our sample (10.2%) as opposed to rates ranging from 31.3% sex work in the last 3 months to 63.9% sex work ever.

### 4.5. SGM-Based Violence and Mental Health

Comparisons of those who did and did not experience SGM-related violence on our various mental health variables revealed associations between violence and some mental health indicators but not others. There were statistically significant differences between those who had experienced SGM-related violence and those who had not with regard to their levels of depressive symptoms. When examining levels of depressive symptoms that warrant therapeutic intervention (moderately severe and severe), those who experienced SGM-related violence had double these rates, as compared to those who reported no SGM-related violence (18.6% vs. 9.2%). This association between SGM violence and elevated levels of depressive symptoms has also been reported in prior studies with GBMSM, both in bivariate and multivariate analyses [5,6]. This is believed to be the first study to demonstrate this in a sample that included both cisgender sexual minority woman and gender minority individuals in Kenya.

PTSD symptoms were found to be significantly higher among those who experienced SGM-related violence. Among those who reported experiencing SGM-related violence, two-thirds (67.6%) reported clinically significant levels of PTSD symptoms, as compared to less than half (42.2%) of those who had not experienced. Psychological distress scores did not vary significantly between those who did and did not experience SGM-related violence, with 13.7% of those who experienced violence reported clinically significant levels of psychological distress, and 10.3% of those who did not experience violence reported clinically significant levels of distress.

### 4.6. Supports and Services Needed

Just over half of the participants (53.3%) responded to an open-ended question at the end of the survey asking respondents to indicate additional support and services that are needed for SGM people in Kenya. The six specific areas of support that they requested are as follows (listed from most to least frequently requested): Counseling and Mental Health Services, Financial and Economic Empowerment, Emotional/Peer Support, Community Organization Support and Advocacy, Healthcare Assistance and Medical Coverage, Sexual Education and Prevention Programs. Of note is that the most commonly provided type of program in Western Kenya for SGM people (primarily GBMSM and transgender women) is Sexual Education and Prevention Programs, and this was the least commonly requested type of service.

These findings provide avenues for future interventions and programs that could be delivered to the SGM community in Kenya. The most requested service (Counseling and Mental Health Services) speaks to the need to develop SGM-sensitive and specific venues for the provision of mental health services to members of the SGM community, as well as to integrate mental health services into existing programs and services for SGM people. The second most requested service (Financial and Economic Empowerment), along with our demographic data which revealed that only 10.6% of the sample currently had full-time employment, speaks to the need for economic development, job skills training, and other employment-focused programs for SGM people. Due to pervasive societal stigma that SGM people experience in Western Kenya, some may experience discrimination in hiring and in the workplace. Support and networking services may assist SGM adults in both acquiring and maintaining employment in venues that are SGM-affirming.

### 4.7. Implications of the Findings for Public Health Practice and Public Policy

In July 2020 the Kenyan Ministry of Health’s Taskforce on Mental Health recommended to the government that mental illness should be declared a National Emergency of epidemic proportions, and urged them to prioritize mental health as a priority public health and socioeconomic agenda [62]. The taskforce report and the Kenya Mental Health Policy 2015–2030 [61] discuss current challenges in promoting mental health in Kenya due to misconceptions and misinformation regarding the origins and course of mental health disorders, and pervasive stigma and discrimination against those living with mental health challenges. Public health and policy interventions for SGM people in Kenya will need to be sensitive to more general societal and cultural understandings and conceptualizations of mental health and mental disorders, while also addressing the unique needs of this marginalized population. It is also important for both researchers and practitioners to be mindful that mental health is a socially constructed and defined concept, thus different cultures and societies may vary in how they view the origins of mental health challenges, how they determine what is mentally healthy and unhealthy, and what interventions they perceive as being culturally appropriate. Findings and recommendations from the current study, as well as those that have preceded it should be viewed through this cautionary lens.

Our data regarding the large number of participants reporting sub-clinical levels of psychological distress and depressive symptoms suggest that SGM people and communities have developed resilience processes that serve to protect them from the deleterious effects of both individual-level and structural-level oppression. Such resilience and protective processes have previously been demonstrated among gay and bisexual young men in Kisumu in a study focused on both sexual and mental health [13]. Future practice efforts aimed at improving the mental health and wellbeing of SGM communities in Kenya should build collaborations with SGM-specific CSOs who work directly with these communities, and have the indigenous knowledge and understanding of how to build on existing resilience processes and coping resources. Additional funding should also be directed toward these CSOs in order to provide needed services for people in safe LGBTQ community settings, by members of the SGM community.

All of these efforts should be designed with an awareness of the diverse ways in which SGM people in Kenya may conceptualize and construct their sexual orientation, gender identity and gender expression. Potential consumers of services may not adhere to Western conceptualizations of sexuality and gender, and resist labels associated with the LGBTQ nomenclature. In addition, mental health prevention and treatment efforts should be cautious about adopting SGM-tailored programs and services developed in non-Kenyan settings, as these may not address the culturally specific needs of clients.

Despite the promising data on general mental health, data on PTSD among SGM people was discouraging, with more than half of SGM adults in our sample reporting clinically significant levels of PTSD symptoms. These findings suggest an urgent need to provide SGM populations with mental health services that address the negative effects of trauma and violence that many SGM people in Kenya experience due to LGBTQ-focused stigma and discrimination. These services should be developed and provided using trauma-informed principles of care, and be sensitive to the unique life circumstances and stressors experienced by SGM people in Kenya. Our open-ended question regarding the needs of SGM people in Kisumu demonstrated that the most frequent request was for Counseling and Mental Health Services. There is currently a dearth of mental health providers and services that support the mental health and wellbeing of SGM people in Kenya. All mental health providers should be trained in therapeutic approaches that are LGBTQ-sensitive, and mental health provider training programs should include modules specific to the needs of SGM communities.

In addition to individual-level therapeutic approaches to improving the mental health and wellbeing of SGM people and communities, SGM-based violence and those who perpetrate it need to be addressed at community, cultural, and policy levels. Nearly half of our sample reported experiencing at least one episode of anti-LGBTQ violence that was perpetrated against them because of their sexual orientation, gender identity, and/or gender expression. In some instances, those in society who should be protecting Kenyan citizens, namely police, healthcare providers, religious leaders, and government officials, perpetrated this violence. Two-thirds of participants who experienced such violence reported clinically significant levels of PTSD, and had double the rates of moderately severe to severe depressive symptoms as compared to those who had never experienced SGM-based violence. These associations support the Minority Stress Model projection of increased mental health challenges among those who experience anti-LGBTQ stigma and discrimination.

Cultural and community interventions that focus on destigmatizing SGM people and communities are needed, including specific sensitization and stigma reduction interventions with various groups of individuals in service sector positions who have been identified as potential perpetrators of violence (e.g., police, healthcare providers, religious leaders). It will be helpful to create spaces for open dialogue and discussion between members of the SGM community and those in various service sectors, in order to demystify sexual orientation and gender identity and to find commonalities across the groups. Such discussions can benefit from taking a human rights approach, and focusing on a shared national identity as Kenyans. Each sector may need additional focus on the intersection of their service provision and the SGM community. For instance, Kenya’s 2010 constitution anchors police reform in that it stipulates that the institution must protect Kenyan’s fundamental rights and freedoms. Highlighting the humanity and fundamental human rights of SGM people in Kenya may encourage police officers to re-conceptualize their role as protecting Kenyan citizens (regardless of sexual orientation or gender identity) and their constitutional rights since the Kenyan constitution states that every person is equal before the law and shall enjoy equal protection by the law. Media-based interventions also may help to reduce SGM-based stigma through the presentation of LGBTQ Kenyans in various social and occupational roles in an effort to normalize SGM people and their lived experiences. Policy related to punishment for SGM-based violence and hate crimes also needs to be institutionalized and enforced in order to prevent future acts of violence.

In order to create sustainable improvements in the mental health and wellbeing of SGM people in Kenya, there needs to be a political level prioritization of LGBTQ populations, and a securing of human rights and services outside of the HIV narrative. The vast majority of health-related policy and resources that involve any segment of the SGM community are currently focused on GBMSM, and more recently transgender women, and their identification as “key populations” for HIV prevention and treatment services. This HIV work often focuses primarily on mechanics of viral transmission and sexual behavior, is devoid of a consideration of cultural factors, and eliminates segments of the larger SGM community such as cisgender sexual minority women and transgender men. It often fails to take a holistic approach to the lives of SGM people, and puts a primary focus on sex and sexuality—topics that are often taboo in Kenyan culture.

Although there has been an increased focus on improving mental health in Kenya, especially with the release of the Kenya Mental Health Policy 2015–2030, there has not been a specific focus on the mental health needs of SGM people in the country. This invisibility of the SGM community in national policy on mental health further restricts critical funding for mental health services that are urgently needed. Greater levels of social integration of SGM people into mainstream society are needed, as well as advocacy and education at all levels around understanding broader definitions of gender, sexuality, and identity. Such efforts may occur with collaboration from LGBTQ-specific CSOs, as well as allies that support affirming and supportive messaging. Allies can also play a critical role in joining LGBTQ activists in fighting for equal rights and protections under the law, such as the decriminalization of same-sex behavior. LGBTQ-inclusive health policies at the local, county, and national levels are also needed to ensure that SGM people have access to needed health services and feel comfortable accessing services in a safe and affirming environment. In order to truly improve the mental health and wellbeing of SGM people in Kenya, multiple and varied efforts will be needed to protect this population from exclusions, restrictions, harm and subsequent mental health challenges.

## 5. Conclusions

The substantial number of participants in this study who reported sub-clinical levels of psychological distress and depressive symptoms in the midst of pervasive anti-SGM stigma and violence in Kenya suggests that SGM people and communities have developed resilience processes that serve to protect them from the deleterious effects of individual-level and structural-level oppression. On the other hand, elevated rates of PTSD suggest an urgent need to develop and deliver culturally appropriate mental health services for SGM adults. These therapeutic interventions should be provided using trauma-informed principles of care, and be sensitive to the lived experiences of SGM adults in Kenya and differences across SOGI categories. Community and policy-level interventions are also needed to decrease SGM-based stigma and violence, increase SGM visibility and acceptance, and create safe and affirming venues for the delivery of both mental and physical health care. In order to create sustainable improvements in the mental health and wellbeing of SGM people in Kenya, there needs to be a political level prioritization of SGM populations, and a securing of human rights and services outside of HIV-focused programs. The Kenya Mental Health Policy 2015–2030 would also benefit from an expansion to address the unique mental health needs of SGM people and communities. In order to truly improve the mental health and wellbeing of SGM people in Kenya, multiple and varied efforts will be needed to protect this population from exclusions, restrictions, harm and subsequent mental health challenges.

## Figures and Tables

**Figure 1 ijerph-18-01311-f001:**
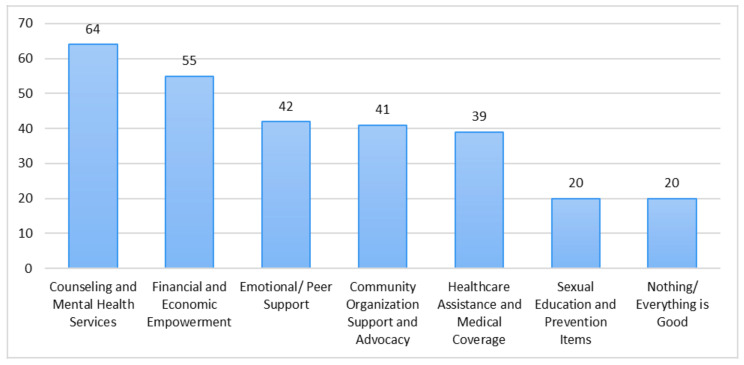
Types of Support and Services Needed.

**Table 1 ijerph-18-01311-t001:** Sample General Demographics.

Demographics	N (%)
Age Groups (*n* = 521) (range 18–54)	
18 to 24	262 (50.3%)
25 to 34	221 (42.4%)
35 or over	38 (7.3%)
County of Residency (*n* = 525)	
Kisumu	305 (58.1%)
Siaya	69 (13.1%)
Bungoma	47 (9%)
Kakamega	34 (6.5%)
Migori	19 (3.6%)
Vihiga	17 (3.2%)
Busia	14 (2.7%)
Kisii	6 (1.1%)
Homabay	4 (0.8%)
Nakuru	3 (0.6%)
Kericho	1 (0.2%)
Other	6 (1.1%)
Religion (*n* = 526)	
Catholic	199 (37.8%)
Anglican	113 (21.5%)
Seventh Day Adventist	78 (14.8%)
Muslim	63 (12.0%)
Indigenous	12 (2.3%)
Other	61 (11.6%)
Employment Status (*n* = 521)	
Part-time work	178 (34.2%)
No work or school	94 (18%)
No work but in school	80 (15.4%)
Full- time work	55 (10.6%)
Sex worker	55 (10.6%)
Laborer	49 (9.4%)
Other or more than one selected	10 (1.9%)
Highest Level of Education (*n* = 526)	
Primary School	61 (11.6%)
Secondary School	310 (58.9%)
Certificate	60 (11.4%)
Diploma	63 (12%)
Bachelor’s Degree	29 (5.5%)
Master’s Degree	3 (0.6%)

**Table 2 ijerph-18-01311-t002:** Sample Sexual Orientation and Gender Identity.

Sexual Orientation and Gender Identity	N (%)
Sexual Orientation (*n* = 527)	
Bisexual	250 (47.4%)
Gay	126 (23.9%)
Lesbian	70 (13.3%)
Men who have sex with men (MSM)	53 (10.1%)
Women who have sex with women (WSW)	11 (2.1%)
Other	9 (1.7%)
Queer	7 (1.3%)
Heterosexual	1 (0.2%)
Gender Identity (*n* = 526)	
Cisgender	464 (88.2%)
Transgender	57 (10.8%)
Gender Non-binary	5 (1.0%)

**Table 3 ijerph-18-01311-t003:** Psychological Distress, PTSD Symptoms, Depressive Symptoms, IPV, SGMV, and Problematic Substance and Alcohol Use: Comparison across Sexual Orientation and Gender Identity (SOGI) groups.

	Cisgender Sexual Minority Women (*n* = 131)	Cisgender Sexual Minority Men (*n* = 336)	Gender Minority Individuals (*n* = 60)	ӽ^2^, *p*
Psychological Distress (BSI-18)				
	% (std resid)	% (std resid)	% (std resid)	
Clinically Significant Levels of Psychological Distress (*t*-score > 62)	9.5% (−0.7)	12.3% (0.3)	12.7% (0.2)	ӽ^2^ = 0.765, *p* = 0.682
Post-Traumatic Stress Disorder (PTSD) Symptoms (PC-PTSD)				
Clinically Significant PTSD Symptoms (symptom total ≥ 3)	50.4% (−0.4)	52.4% (−0.2)	64.3% (1.1)	ӽ^2^ = 3.257, *p* = 0.196
Depressive Symptoms (PHQ-9)				
Clinically Significant Levels of Depressive Symptoms (score ≥ 10)	22.5% (−0.8)	28.4% (0.8)	21.4% (−0.7)	ӽ^2^ = 2.372 *p* = 0.305
Intimate Partner Violence (IPV)				
EVER experienced Intimate Partner Violence	43.8% (0.1)	38.4% (−1.3)	69.9% (3.0 *)	ӽ^2^ = 19.061 *p* = 0.000 **
Sexual/Gender Minority-Based Violence (SGMV)				
EVER experienced Sexual/Gender Minority-Based Violence	37.7% (−1.0)	38.3% (−1.4)	84.2% (4.7 *)	ӽ^2^ = 43.917 *p* = 0.000 **
Kenya NACADA Household Questionnaire (Those who had DAILY usage)				
Tobacco	10.9% (−1.7)	21.4% (1.9)	5.6% (−2.0 *)	ӽ^2^ = 12.78, *p* = 0.002 **
Alcohol	15.5% (−0.4)	19.4% (1.1)	5.5% (−2.1 *)	ӽ^2^ = 6.74, *p* = 0.034 **
Marijuana	4.7% (−1.9)	13.3% (2.0 *)	1.8% (−1.9)	ӽ^2^ = 12.10, *p* = 0.002 **
Home Brew	3.1% (−1.0)	6.5% (1.1)	1.9% (−1.0)	ӽ^2^ = 3.50, *p* = 0.174
Miraa or Khat	0% (−1.0)	1.3% (0.9)	0% (−0.7)	ӽ^2^ = 2.28, *p* = 0.320

* indicates residual strengths (+/− 2 values; higher or lower than expected vs. observed counts) between SOGI groups. ** indicates *p* value statistical differences between groups with an alpha of 0.05.

**Table 4 ijerph-18-01311-t004:** Psychological Distress, PTSD Symptoms and Depressive Symptoms: Comparisons with Lifetime Experiences of Sexual/Gender Minority-Based Violence.

	Never Experienced SGMV (*n* = 296)	Ever Experienced SGMV (*n* = 225)	ӽ2, *p*
	% (std resid)	% (std resid)	
Depressive Symptoms (PHQ-9)			ӽ^2^ = 18.07, *p* = 0.000 *
Clinically Significant Levels of Depressive Symptoms (score ≥ 10)	18.6% (−2.4)	35.0% (2.8)	
PTSD Symptoms (PC-PTSD)			ӽ^2^ = 32.72, *p* = 0.000 *
Clinically Significant PTSD Symptoms (symptom total > 3)	41.4% (−2.6)	67.3% (3.1)	
Psychological Distress (BSI-18)			χ^2^ = 1.57, *p* = 0.210
Clinically Significant Levels of Psychological Distress (*t*-score > 62)	9.9% (−0.8)	13.5% (0.9)	

* indicates *p* value statistical differences between groups with an alpha of 0.05.3.5. Types of Support and Services Desired.

## Data Availability

Restrictions apply to the availability of these data. Data were obtained from NYARWEK LGBTI Coalition and are available from the authors with permission from the Executive Director of NYARWEK LGBTI Coalition.

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
