# Peer review of "Mental Health Challenges and Needs among Sexual and Gender Minority People in Western Kenya"

_ijerph, 2021, doi:10.3390/ijerph18031311_

Round 1

Reviewer 1 Report

The authors present an important topic, whose exploration is greatly warranted. Great work

Some background/discussion enhancement is needed:

Non-heteronormative populations have always existed on the continent of Africa – their presence is not new. The legacy of colonialism and the adoption of Eurocentric-Western religiosity has stigmatized these groups who many now be "remerging". This reality is foundational to the experiences on the content and current criminalizing laws are based on this reality. The background and/or discussion needs to reflect this.

Given that non-heteronormative person have always existed and were accepted in pre-colonial societies, paradoxically, there is also a need to appreciate that the term LGBTQ – is Eurocentric-western in origin. A sample of the question used to demonstrate participants self-identified as LGBTQ would be useful. Were there other options given in language to solicit such identities?

Depending, it needs to be appreciated that not using these terms does not indicate a denial of who people are, but that LGBTQ and other Western-based language in this part of the world may be perceived different. Or simply people may have a different way of self-conceptualizing. Understanding this more can be intervention goals but not appreciating this may contribute to societal/cultural/and government resistance to address the needs of this population. Same goes for the concept of gender. Literature on this may be more prevalent in the West African context, but the this is also seen among U.S. Black male populations (i.e. use of the term same gender loving).

Expound on comparing the rates of childhood abuse, drug abuse depression and PTSD among this sample to the national rates.

Some more discussion is needed on the cultural relevancy of the Minority Stress Framework, as well as the tools used. Are the Cronbach alphas available for the tools for this sample. A table with this information or text is warranted. A limitation is the use of these constructs of mental health without discussion of cultural relevancy of the term mental distress and related concepts. The low proportion of those endorsing psych distress may reflect this or not, given national prevalence.

According to the Table 90% of the sample reported not having full-time employment. This is a VERY large proportion and warrants more discussion, and may be reflected in support and services reportedly desired. This is also reflective of a larger societal problem where intervention is needed. Any insight on what proportion of this is contributing to mental distress? How does this compare nationally and for the female sex in the country?

The discussion on violence is important. Given that state-sanctioned violence by police is a global phenomenon (in the U.S. it is based on race) , it is important to contextualize this a bit more. What is the level of violence against other marginalized groups (i.e. tribalism is still an issue, female/women) by police/law enforcement. Intervention is need in re-imagining policing overall, very brief discussion how could this better society could enrich discussion. The IPV discussion needs more development, what proportion were perpetuators as well as victims? At times there is great overlap here. What are potential individualized/interpersonal interventions that warrant consideration?

Overall, this is a good report. But enhancing the conceptual and contextual elements is warranted.

Reviewer 2 Report

This study examines psychological distress, depressive symptoms, PTSD symptoms, and alcohol and substance use among 521 Kenyan who identified as sexual and gender minority (GCM).

The authors mentioned “minority stress framework” and used it as their major theoretical rationale for the study; however, no definition was given for the framework. The literature review mostly discussed the situation in Kenya. Explaining the situation in Kenya is important, but the discussion of theory and relevant existing studies around the world are equally important. I recommend that the authors add at least a paragraph or two explaining their theoretical rationale, such as the minority stress framework and research on mental health outcomes from Kenya or other parts of the world. This will allow readers to understand the situation and the existing research thoroughly.

The authors mentioned the sample sizes and sample characteristics in results; however, they should belong to section 2.1 Sampling. Their results should be relating to their research questions, i.e. the mental health symptoms of GCM in Kenya, not the sample characteristics. Sample characteristics are part of the method (i.e. telling readers how they sampled and the distribution of characteristics of their sample)

Also, the authors need to provide more details on their sampling method.  They mentioned recruiting through social media. Did they use any social media groups/pages to recruit? They said by “word of mouth” – but who invited the participants? All these will tell us whether the sample is biased in one way or the other (e.g., if they recruit, say, through a university page, then we can assume that the participants are biased towards higher education or at least, people who are interested in university education).  These details, although appear to be minimal provide important details to readers on the sampling method. This will also affect how future researchers replicate the study.

Please explain if any of the materials were translated to other language or presented as is (in English). If translation was used, please explain if back translation was used to increase validity of the translation.

Reliability (cronbach’s alphas) should be reported individually per measurement, rather than as a summary in line 200.

The presentation of the results  in 3.3 onwards is confusing. The authors mentioned using chi-square AND one-way ANOVA. But the results all showed chi-squares and p, rather than F and p. Please clarify what tests were done for each of the analysis. T-tests or One-way ANVOA should be more appropriate than chi-square tests, since scores for symptoms are continuous or at least, ordinal. However, given that the study has unequal sample sizes among groups (e.g., 131 cisgender sexual minority women, 336 cisgender sexual minority men, and 60 gender minority individuals); therefore, the assumption for equal variance in one-way ANOVA is likely violated. The authors should consider non-parametric tests.  Regardless of what the authors end up choosing, the authors need to explain clearly their analytic rationale in section 2.7.

Minor issues:

Line 47 - “…an increasingly more” – reduce redundancy by deleting “more”. Increasing means more.
